# Phytochemical composition and bioactivity of *Debregeasia saeneb* leaves: Insights into anti-diabetic and antioxidant properties

Rashid Khan[1], Rabia Afza[1]*, Bashir Ahmad [2,3¤]*, Sumaira Miskeen[4], Khalid Ahmad[5], Mostafa A. Abdel-Maksoud [6], Salman Alrokayan[6], Mohamed Y. Zaky[7]

1 Department of Botany, Hazara University Mansehra, Khyber Pakhtunkhwa, Pakistan, 2 Department of Pediatrics, Affiliated Hospital of Guangdong Medical University, Zhanjiang, China, 3 Department of Biology, The University of Haripur, Khyber Pakhtunkhwa, Pakistan, 4 Department of Food Science and Technology, The University of Haripur, Khyber Pakhtunkhwa, Pakistan, 5 Department of Environmental Sciences, COMSATS University Islamabad, Abbottabad Campus, Abbottabad, KPK Pakistan, 6 Chair of Biomedical Applications of Nanomaterials, Biochemistry Department- College of Science- King Saud University, Riyadh, Saudi Arabia, 7 Molecular Physiology Division, Zoology Department, Faculty of Science, Beni-Suef University, Beni-Suef, Egypt

¤Current address: Department of Pediatrics, Affiliated Hospital of Guangdong Medical University, 524000 Zhanjiang, China
* ethnopk@gmail.com (RA); bashir18840@gmail.com (BA)

## Abstract

Medicinal plants are an essential reservoir of natural compounds with diverse pharmacological properties. This study focuses on *Debregeasia saeneb,* (Forsk.) Hepper & J.R.I.Wood a relatively unexplored plant recognized for its traditional medicinal uses. The research aims to identify and analyze the phytochemical composition of *D. saeneb* (Forsk.) Hepper & J.R.I.Wood leaves, as well as evaluating their antioxidant and antidiabetic potential. This study systematically identified and analyzed the phytochemical constituents of *D. saeneb* (Forsk.) Hepper & J.R.I.Wood leaves utilizing Gas Chromatography-Mass Spectrometry (GC-MS), alongside both quantitative and qualitative assays. The biological activities were assessed through radical scavenging assays, specifically DPPH (1,1-di-phenyl-2-picrylhydrazyl), ABTS (2,2-azino-bis(3-ethylbenzothiazoline-6-sulfonic acid)), and α-amylase inhibitory assays, which have been largely underexplored across various solvents. The GC-MS analysis revealed a diverse range of compounds, with ethyl acetate extracts containing six compounds, chloroform extracts containing seven, and n-hexane extracts containing three distinct compounds. Further qualitative assessments confirmed the presence of glycosides, leucoanthocyanins, quinones, lignins, carbohydrates, terpenoids, anthraquinones, and phlobatannins in the extracts of *D. saeneb* (Forsk.) Hepper & J.R.I.Wood. Quantitative analyses established the total phenolic, tannin, and flavonoid contents, revealing a positive correlation among the different extracts. Specifically, the total flavonoid (TF) content in the standard ranged from 0.34 to 1.189 µg/mL, while in methanol extracts it ranged from 0.087 to 0.778 µg/mL, in ethyl

**Data availability statement:** The data inculed in this manuscript were generated during experiment and the method and introduction from other papers are cited accordingly.

**Funding:** We are very thankful to Higher Education of Pakistan for financial support (grant reference number: 533/IPFP-II(Batch-I)/NAHE/HEC/2020/126). The authors extend their appreciation to the Researchers Supporting Project number (RSPD2023R725) King Saud University, Riyadh, Saud Arabia. The funders had no role in study design, data collection and analysis, decision to publish, or preparation of the manuscript.

**Competing interests:** The authors have declared that no competing interests exist.

acetate from 0.188 to 0.624 µg/mL, and in n-hexane from 0.03 to 0.44 µg/mL. The total tannin content (TTC) in the standard ranged from 0.462 to 2.359 µg/mL, with methanol extracts showing values from 0.016 to 0.048 µg/mL, ethyl acetate from 0.0196 to 0.153 µg/mL, and n-hexane from 0.0012 to 0.134 µg/mL. The total phenolic content (TPC) in the standard ranged from 0.11 to 0.38 µg/mL, with methanol extracts ranging from 0.042 to 0.267 µg/mL, ethyl acetate from 0.0275 to 0.487 µg/mL, and n-hexane from 0.001 to 0.348 µg/mL. In ethanol extracts, the radical scavenging capacity (RSC) ranged from 0.005 to 0.05 µg/mL, in methanol from 0.01 to 0.09 µg/mL, and in aqueous extracts from 0.005 to 0.035 µg/mL, while the standard RSC ranged from 0.5 to 4.5 µg/mL. The biological activity assays indicated that *D. saeneb* (Forsk.) Hepper & J.R.I.Wood extracts exhibited significant α-amylase inhibition, suggesting potential anti-diabetic properties. Furthermore, all extracts demonstrated radical scavenging activity in both ABTS and DPPH assays, with the methanol extract exhibiting the highest antioxidant activity. In conclusion, this research provides comprehensive insights into the phytochemical profile of *D. saeneb* (Forsk.) Hepper & J.R.I.Wood leaves and its potential therapeutic applications, including anti-diabetic and antioxidant effects, thereby warranting further investigation into molecular mechanisms, drug development, and health promotion.

## Introduction

Medicinal flora are a primary source of natural drugs for the treatment of various illnesses, and their popularity has increased over time alongside growing knowledge about plants [1–3]. Researchers are now interested in exploring different plants for new, safe, and effective drugs to combat a range of diseases [4–6]. These drugs are derived from various extracts and isolated compounds; consequently, different techniques are employed to isolate these phytochemicals from plants [7]. A plethora of research reveals that plant secondary metabolites have significant potential for the treatment of various diseases, including cancer, and diabetes [6,8–12]. In plants, secondary metabolites include terpenoids, phenols, flavonoids, tannins, and saponins, among others [8]. *Debregeasia saeneb* (Forsk.) Hepper & J.R.I.Wood (Syn: salicifolia (D. Don) Rendle) is also recognized as [13]. *D. saeneb* belongs (Forsk.) Hepper & J.R.I.Wood to the family Urticaceae. Its leaves are oblong, the fruits are globose and yellow, and flowers are clustered and sessile [14]. According to one report, in which the authors qualitatively screened the phytochemicals in methanol extracts, the *D. saeneb* (Forsk.) Hepper & J.R.I.Wood leaf extract contain flavonoids, anthocyanins, tannins, anthraquinones, alkaloids, and saponins [15]. Another study demonstrated that *D. saeneb* (Forsk.) Hepper & J.R.I.Wood may contain various compounds, suggesting its potential utility in the treatment of cancer and gastrointestinal illnesses [16]. Ethnobotanical surveys reveal that *D. saeneb* (Forsk.) Hepper & J.R.I.Wood is traditionally used to treat a range of ailments, including dermatitis, carbuncles, boils, bone fractures, skin rashes, eczema, pimples, tumors, and urinary tract diseases [17]. These therapeutic applications may be due to their antioxidant properties and other beneficial effects.

Oxidation generate free radicals within cells, initiating a series of chemical reactions that lead to cellular damage and contribute to various diseases, including aging, rheumatic diseases, metabolic disorders, diabetes, and cardiovascular diseases [18]. Therefore, it is essential to inhibit unnecessary oxidation in living organisms to prevent these conditions. Plant-derived antioxidants serve as a primary source of disease prevention by mitigating oxidative stress [19]. Oxidation is linked diabetes, which has two major types: Type I and Type II [20]. In Type I Diabetes Mellitus (TIDM), the pancreas fails to produce insulin; consequently, its treatment involves insulin administration and accounts for 10% of all diabetes cases. In Type II Diabetes Mellitus (TIIDM), cells exhibit resistance to insulin, constituting 90% of all diabetes cases, and are treated with either synthetic or natural medications to overcome this resistance [21,22]. The enzyme alpha-amylase facilitates the digestion of oligosaccharides in the intestine, breaking them down into absorbable monosaccharide in the small intestine. These monosaccharides are subsequently absorbed into the bloodstream, resulting in elevating the glucose levels. In individuals with diabetes, rising blood glucose can lead to various complications within the body [23]. Consequently, the inhibition of alpha-amylase is a primary strategy for reducing blood glucose levels during diabetes by preventing the digestion of carbohydrates in the gastrointestinal tract (GIT) into simple sugars. Medicinal plants contain a variety of phytochemicals that can influence the activity of the alpha-amylase enzyme. Phenolic compounds, including phenolic acids and flavonoids, interact with alpha-amylase and modify its activity [24].

Medicinal plants are valuable sources of bioactive compounds with therapeutic potential. Despite their traditional medicinal uses and ethnobotanical significance, *Debregeasia saeneb* (Forsk.) Hepper & J.R.I.Wood remains underexplored, particularly regarding its phytochemical composition and biological activities. Oxidative stress and dysregulation of carbohydrate metabolism are central to the pathogenesis of chronic diseases such as diabetes, highlighting the need for novel antioxidants and anti-diabetic agents. This study aims to address this gap by comprehensively analyzing the phytochemical profile of *D. saeneb* (Forsk.) Hepper & J.R.I.Wood using advanced techniques such as gas chromatography-mass spectrometry (GC-MS) and evaluating its antioxidant and α-amylase inhibitory activities across different solvent systems. These investigations not only enhance our understanding of the therapeutic potential of *D. saeneb* (Forsk.) Hepper & J.R.I.Wood but also provide a scientific basis for its traditional medicinal uses and pave the way for further molecular and pharmacological studies.

## Materials and methods

### Plant collection and identification

Specimens of *D. saenab* (Forsk.) Hepper & J.R.I.Wood were collected from multiple sampling points within the Ghazi Tehsil region, located in District Haripur, Khyber Pakhtunkhwa (KPK), Pakistan. The collection sites were chosen based on the natural distribution and abundance of the plant within the habitat. Geographical coordinates and altitude for each sampling point were recorded using a handheld GPS device to ensure accurate documentation of collection locations. Habitat observations, including soil type, vegetation composition, and general ecological conditions, were noted during field surveys. Specimens were collected with their complete vegetative and reproductive parts to facilitate proper identification and herbarium preparation. Each sample was assigned a unique collector number for reference. The collector numbers were labeled on-site and cross-referenced with their respective GPS coordinates for further analysis. All collected specimens were placed in herbarium sheets and processed according to standard botanical protocols for preservation and taxonomic study.

Following collection of *D. saeneb*, (Forsk.) Hepper & J.R.I.Wood the plant was identified through e-flora of Pakistan and consultation with an expert botanist in the Department of Biology (Botany) at the University of Haripur, Pakistan. Following authentication, the leaves of *D. saeneb* (Forsk.) Hepper & J.R.I.Wood were gently washed with distilled water to remove debris and air-dried under shade at room temperature (20–30°C) with adequate air circulation to prevent mold growth. The leaves were spread evenly on non-porous paper to facilitate uniform drying. Direct sunlight was avoided to preserve bioactive compounds. After drying, the leaves were pulverized into a powder and sieved through a 60-mesh stainless steel sieve. The Department of Botany at the University of Haripur, Pakistan, received the voucher specimens for deposit.

## Crude extract preparation from the dried powder

Crude extracts from leaves in different solvents (Aqueous, methanol, ethanol, ethyl acetate, and n-hexane) were obtained following maceration method as we have described previously [9,25]. The extraction was carried out in the following order of solvents based on their polarity: aqueous extraction (highly polar), followed by methanol, ethanol (both polar), then ethyl acetate (semi-polar), and finally n-hexane (non-polar) or from polar to non-polar. In summary, 200 grams of powdered leaves were added to 500 ml of solvent and allowed to soak for five days. After soaking, the mixtures were filtered using Whatman No. 1 filter paper. The filtrates were then concentrated with a rotary evaporator at 40 ºC under low pressure. The resulting crude extracts were labeled according to the plant name and stored at 4 ºC for future use. The extraction method is illustrated in Fig 1.

## Sample preparation for GC-MS analysis

The *D. saeneb* (Forsk.) Hepper & J.R.I.Wood leaves powder (25 g) was soaked in 95% n-hexane, chloroform, and ethyl acetate for 5 days. Sediments from the filtrates were removed using Whatman filter No. 1 along with 2 g of sodium sulfate. Prior to filtration, the filter paper was moistened with a solution of sodium phosphate and 95% ethanol to minimize the risk of plant extract components, such as phenolics, flavonoids, or enzymes, adhering to the dry paper and being lost during the filtration process. After filtration, the filtrates were concentrated using nitrogen injections, and 2 µL of the solution was used for GC-MS analysis.

## GC-MS analysis and components identification

The GC-MS analysis of leaf extracts was conducted using the Agilent 5975C GC-MS system, which consisted of an auto-sampler and GC-MS apparatus. The setup featured an elite 1 attached Slica column (300.25 mmID × 1EM df, Dimethyl Polysiloxane 100%). It operated in Electron Impact Mode at 70 eV. Helium gas (99.99%) served as the carrier gas, maintaining a flow rate of 1.51 mL/min. The injector temperature was set at 1 µL volume, while the ion source temperature was maintained at 200 ºC. The temperature control was automated to increase from 70 ºC to 300 ºC at a rate of 10 ºC per minute, concluding with a hold at 300 ºC for 9 minutes. Additionally, mass spectra were recorded in the scan range of 40–1000 m/z at 70 eV. After 2 minutes, the solvent flow was halted, and mass spectrometry commenced at 2.96 minutes, concluded at 16.5 min. The components were identified through the analysis of molecular structures, predicted fragments, and molecular masses. The GC-MS data were analyzed using the mass spectrometry interpretation component of the NIST database, which contains over 62,000 distinct patterns. The names, structures, and molecular weights of the tested materials were identified, and the relative percentages of each component were determined by comparing peak areas to the total area. Subsequently, the unknown spectra of the tested material were compared to spectra in Turbomass 5.2 software, utilizing the NIST library version (2005) [9,25].

## Qualitative and quantitative analysis of phytochemicals of *D. saeneb* (Forsk.) Hepper & J.R.I.Wood leaves extracts

**Qualitative analysis of different compounds.** Different compounds in two extracts of *D. saeneb* (Forsk.) Hepper & J.R.I.Wood leaves were qualitatively analyzed using the following tests.

**Sulfuric acid test:** Glycosides in leaf extracts were evaluated using a modified version of previously described methods [25]. In summary, 0.5 mL leaves extract was combined with 1 mL of concentrated H2SO4, thoroughly shaken, and allowed to stand undisturbed for 2 minutes. The appearance of a reddish-brown color indicated the presence of glycosides .

**Amino acid test (Ninhydrin test):** The method described previously [25] was utilized to assess amino acids in plant extracts. In a 0.5 mL sample of the plant extract, three drops of ninhydrin solution were added and heated in a water bath for 10 minutes. The development of a purple color indicated the presence of amino acids in the leaf extracts.

 

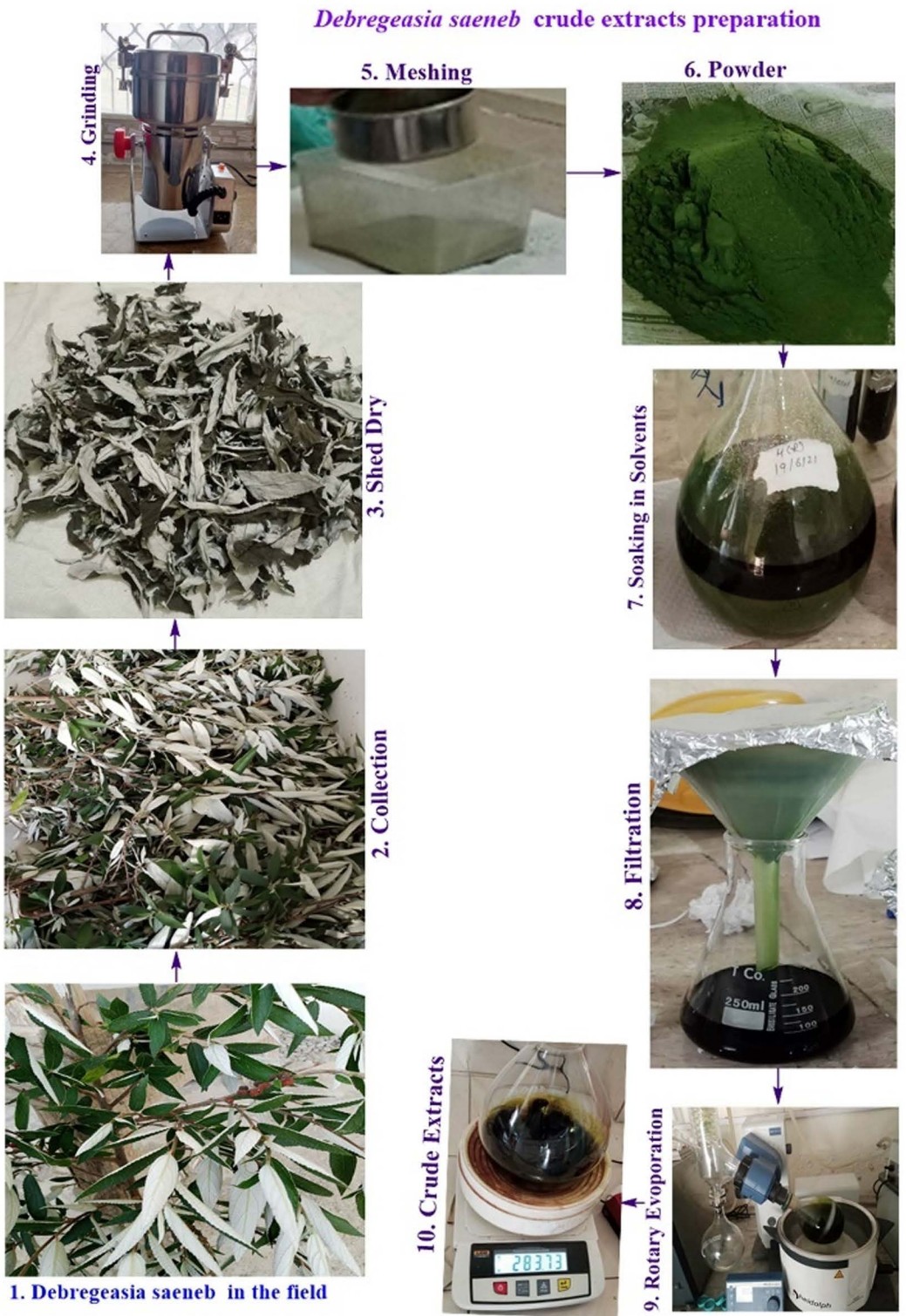

**Fig 1. Sequential extraction process for preparation of crude extracts from *Debregeasia saeneb* (Forsk.) Hepper & J.R.I.Wood leaves: (1) plant collection, (2) shade drying, (3) grinding, (4) sieving, (5) solvent extraction, (6) filtration, and (7) rotary evaporation. All images represent original documentation of experimental procedures.**

**Testing carbohydrates using benedicts test:** The presence of carbohydrates (reducing sugars) in leaf extracts was determined using a previously established method [25]. In 0.5 mL leaves extracts, Benedicts reagent (0.5 mL) was added and then heated in a water bath for 5 minutes. The formation of brick-red precipitates indicated the presence of reducing sugars. Dextrose served as the positive control during the experiment.

**Anthocyanin test:** Anthocyanins in leaf extracts were assessed using the previously established method [25]. In a 200 µL sample of the plant extract, hydrochloric acid (HCl) and 200 µL of ammonia (NH3) were added. A color change from pink-red to blue-violet was observed, indicating the presence of anthocyanins.

**Test for leucoanthocyanins:** The previously employed method [25] was utilized for the detection of leucoanthocyanins with some modifications. In brief, a total of 400 µL of the *D. saeneb* (Forsk.) Hepper & J.R.I.Wood sample was combined with 400 µL iso-amyl alcohol. The presence of leucoanthocyanins was indicated by the observation of a red color in the top layer.

**Emodins test:** The previously described method [25] was employed for checking emodin in plant extracts, with some modifications. In brief, 400 µL of *D. saeneb* (Forsk.) Hepper & J.R.I.Wood sample, 800 µL of ammonium hydroxide, and 1.2 mL of benzene were mixed. The presence of emodin was indicated by the observation of a red color in the ammonia-cal layer.

**Test for anthraquinones:** The previously employed method [25] was utilized for the detection of anthraquinones. In brief, 200 µL of the plant sample, 200 µL of benzene, and ammonia (NH3) were mixed. The presence of violet, pink, and red ammoniacal layers was observed, indicating the presence of anthraquinones in the plant samples.

**Test for terpenoids:** Terpenoids in plant samples were detected using a previously established method [25]. In the leaf extract (500 µL), a few drops of $H_2SO_4$ and chloroform (2 mL) were added. The presence of terpenoids was indicated by the appearance of a reddish-brown color at the interface.

**Quinone test:** Quinones in plant samples were determined using the methods described previously [25]. A 0.5 mL aliquot of the plant sample was combined with 0.5 mL of KOH, mixed thoroughly, and the color spectrum ranging from red to blue was observed for the presence of quinones.

**Checking of lactones through legal test:** The lactones in plant extracts were analyzed using the method as described previously [25]. In plant extracts (4 mg), pyridine (0.5 mL) was added. Following mixing, the mixtures were treated with NaOH and sodium nitroprusside. The appearance of a deep red color indicated the presence of lactones in the leaf sample.

**Lignin tests:** *Labat test:* Lignin in plant samples was analyzed using the methods as previously described [25]. In a 0.5 mL plant sample, 0.5 mL of gallic acid was added and mixed, resulting in the observation of olive-green color indicating the presence of lignin.

**Dahlmann test:** The method previously described [9,25] was employed to determine lignin content in plant samples, with some modifications. Briefly, two drops of aniline and a few drops of concentrated H2SO4 were added to the plant sample (0.5 mL), and the appearance of a yellow color was observed for the presence of lignin.

**Test for phlobatannins:** The method previously described in reference [9,25] was employed to determine the presence of phlobatannins in plant samples. A 0.5 mL aliquot of the plant sample was mixed with concentrated hydrochloric acid (HCl) and heated in water bath for 10 minutes. After boiling, the formation of red precipitates was observed for the presence of phlobatannins.

## Quantitative evaluation

**Quantification of total tannin content (TTC).** The Folin-Cio-Calteu method was employed to determine the total tannin concentration, as previously described [9,25]. Briefly, in a 2 mL Eppendorf tube, 0.75 µL of distilled water, 0.5 µL of Folin-Ciocalteu phenol reagent, 0.1 mL of a 35% $Na_2CO_3$ solution, and 0.1 mL of the plant extract were combined. This mixture was stirred at 30 °C for 30 minutes. Gallic acid solutions at various concentrations (125, 250, and 500 µg/

ml) were used as standards. The absorbance of the blank, test, and standard solutions was checked using a UV-visible spectrophotometer at a wavelength of 725 nm. The total tannin content of the extracts was expressed in mg of GAE/g.

**Measurement of total phenolic content.** The total phenolic content was assessed using the Folin-Ciocalteu method, as previously described [9,25]. In summary, 100 µL of plant extract was combined with 540 µL of distilled water and 60 µL of Folin-Ciocalteu reagent in an Eppendorf tube. The mixture was thoroughly shaken and then incubated at room temperature. After incubation, 600 µL of a 7% $Na_2CO_3$ solution was added, and the mixture was incubated at 30°C for 90 minutes. Subsequently, the absorbance of both the samples and standards was measured using a UV-visible spectrophotometer. The total phenolic content (TPC) was expressed in terms of mg of gallic acid equivalents (GAE) per gram. Standards were prepared using various concentrations of gallic acid (125, 250, and 500 µg/mL).

**Quantification of total flavonoid content (TFC).** The total flavonoid content (TFC) was assessed using a previously described method [9,25]. A stock solution of the plant extract was prepared at a concentration of 5 mg/mL. For the assay, 25 µL, 50 µL, and 100 µL of the plant extract were each mixed with 300 µL of methanol. Subsequently, 20 µL of aluminum chloride ($AlCl_3$) and 20 µL of potassium acetate were added sequentially, followed by the addition of 660 µL of distilled water. After incubating for 30 minutes, the absorbance was recorded at 415 nm using a UV-VIS spectrophotometer (Shimadzu UV-1800). Quercetin dihydrate was used as the positive control, while methanol served as the negative control.

**Quantification of reducing sugar content (RSC).** The reducing sugar content (RSC) was measured using the 3,5-dinitrosalicylic acid (DNSA) method, as previously described [25]. Briefly, in a test tube, 1 mL of the plant extracts were mixed with 2 mL DNSA reagent and incubated for 5 minutes at 95 °C. After incubation, the solution was cooled and mixed with 7 mL of distilled water. The RSC was measured using a UV-VIS spectrophotometer (Shimadzu UV-1800) at a wavelength of 540 nm. The results were expressed as mg D-glucose equivalent/gm of dry weight of the extracts. The reduced sugar content was determined using a standard D-glucose calibration curve (200–1000 mg/L).

### Alpha-amylase inhibitory activities of *D. saeneb* (Forsk.) Hepper & J.R.I.Wood

The alpha-amylase inhibitory activities of *D. saeneb* (Forsk.) Hepper & J.R.I.Wood were evaluated using extracts of methanol, ethyl acetate, and n-hexane, following the method described in [9,25]. In summary, 160 µL of the test solution and enzyme solution were incubated at 37°C for 30 minutes. After incubation, 60 µL of phosphate buffer and 80 µL of starch solution were added, followed by 400 µL of DNS solution. The mixture was then heated at 85-90°C for 15 minutes, cooled, and diluted with 5 mL of water. Absorbance was measured at 540 nm. For the blank, the DNS solution was added prior to the starch solution. For the control, DMSO was used instead of the plant extract, and the acarbose served as standard. The percentage inhibition was calculated using the formula below.

$$\% \ Inhibition = \frac{control - sample}{control} \times 100$$

### Diphenyl picryl hydrazal (DPPH) assay

The DPPH scavenging assay was performed according to a previously described method [25]. Briefly, stock solutions of all plant extracts were prepared at a concentration of 10 mg/mL in methanol. Final concentrations of the plant extracts (125 µg/ml, 250 µg/ml, and 500 µg/ml) were then prepared from the stock solution in a total volume of 100 µl. In 96-well plates, 100 µl of each concentration was combined with 100 µl of a 0.04% DPPH solution in methanol, with each concentration tested in triplicate. The samples were incubated for 30 minutes, after which the absorbance was measured using a spectrophotometer at 517 nm. Methanol alone, without any plant extract served as a blank, while ascorbic acid was utilized as the standard.

## ABTS radical scavenging assay

To evaluate the capacity of plant extracts to neutralize free radicals, the ABTS radical cation (ABTS•+) assay was employed, following the previously used methodology [9,25]. The process commenced by dissolving 5 mg of ABTS, sourced from Glentham Life Sciences in the UK, in 7 mL of phosphate-buffered saline (PBS) from Sigma-Aldrich, aiming for a final concentration of 0.7 mM ABTS. Simultaneously, potassium persulfate from VWR Chemicals BDH Prolabo was used to prepare a 2.45 mM solution in distilled water. Equal volumes of the ABTS and potassium persulfate solutions were combined in a glass container and stored away from light at a stable room temperature of 24 ± 1 °C to facilitate the formation of the ABTS radical. The antioxidant potential was assessed immediately after the ABTS had been oxidized. Test samples were prepared at varying concentrations in DMSO, and 50 μL from each sample was transferred to a 96-well plate. Subsequently, 150 μL of the ABTS mixture was added to each well, and the plate was then allowed to stand at room temperature for 10 minutes. After incubation, a BioTek Synergy HT plate reader was employed to measure the absorbance at 734 nm. A control sample consisting solely of DMSO and the ABTS mixture was also prepared. The formula for calculating the ABTS scavenging activity is as follows:

$$Radical\ Scavenging\ (\%) = \frac{A_{control} - B_{sample}}{A_{control}} \times 100$$

Where A-control is the absorbance of blank and B is the absorbance of samples.

## Statistical analysis

The experiments were conducted in triplicate. Data was analyzed using Excel for correlation and two-way analysis of variance (two-way ANOVA) through SPSS (version 16).

## Results

### GC-MS detection of various phytochemicals in *D. saeneb* (Forsk.) Hepper & J.R.I.Wood leaf extracts

GC-MS is an effective method for identifying volatile compounds in various mixtures, including plant extracts. In the current study, compounds were detected in three different extracts (Ethyl acetate, Chloroform, n-hexane) of *D. saeneb* (Forsk.) Hepper & J.R.I.Wood leaves. In the compound detection procedure, the solvent cut time was approximately 2 minutes, beginning at 2.4 minutes when peaks were first observed and concluding at 16.4 minutes when the last peak was recorded, as shown in Fig 2. The results indicate that approximately 14 small molecules were identified across the three extracts in *D. saeneb* (Forsk.) Hepper & J.R.I.Wood —six in ethyl acetate, seven in chloroform extract, and three in n-Hexane extract. The analysis of ethyl acetate extracts revealed the presence of several compounds, including Heptyl hexacosyl ether, which was detected at 2.164 minutes with a relative abundance of 5.0970%. Hentriacontane was identified at multiple retention times: 2.324, 2.825, 3.36, 3.69, and 3.76 minutes, with corresponding abundances of 8.8499%, 6.8932%, 24.497%, 1.7092%, and 2.6133%. Additionally, Carbonic acid, 2-ethylhexyl nonyl ester was observed at 2.594 minutes with an abundance of 17.815%. Other compounds included Silane, cyclohexyldimethoxymethyl- at 3.145 minutes, Eicosyl heptyl ether at 3.42 minutes (1.4271%), and 1-decanol and 2-hexyl- at 4.43 minutes (4.8028%). In the chloroform extracts, the following compounds were detected: 2,4-di-tert-butylphenol at 4.8 minutes (5.02084%), Dotriacontane at 5.156 minutes (1.22223%), Triacontane at 5.576 minutes (2.30463%), Phytyl, 2-methylbutanoate at 6.156 minutes (6.23625%), Eicosanoic acid at 6.866 minutes (4.04811%), Phytol at 7.612 minutes (5.7720%), and Tetrapentacontane at 6.481 minutes (1.5249%). Furthermore, in the n-hexane extracts, Benzene, 1-ethyl-3-methyl- was detected at 2.209 minutes with a relative abundance of 22.467%. Carbonic acid, decyl undecyl ester was identified at 2.334, 2.825, and 3.335 minutes with an abundance of 11.839%. Hentriacontane was also detected at 3.83, 4.765, and 5.701 minutes, with a total abundance of 22.827%. The results of the gas chromatography-mass spectrometry (GC-MS) analysis are further summarized in Figs 2 and 3, as well as in Table 1.

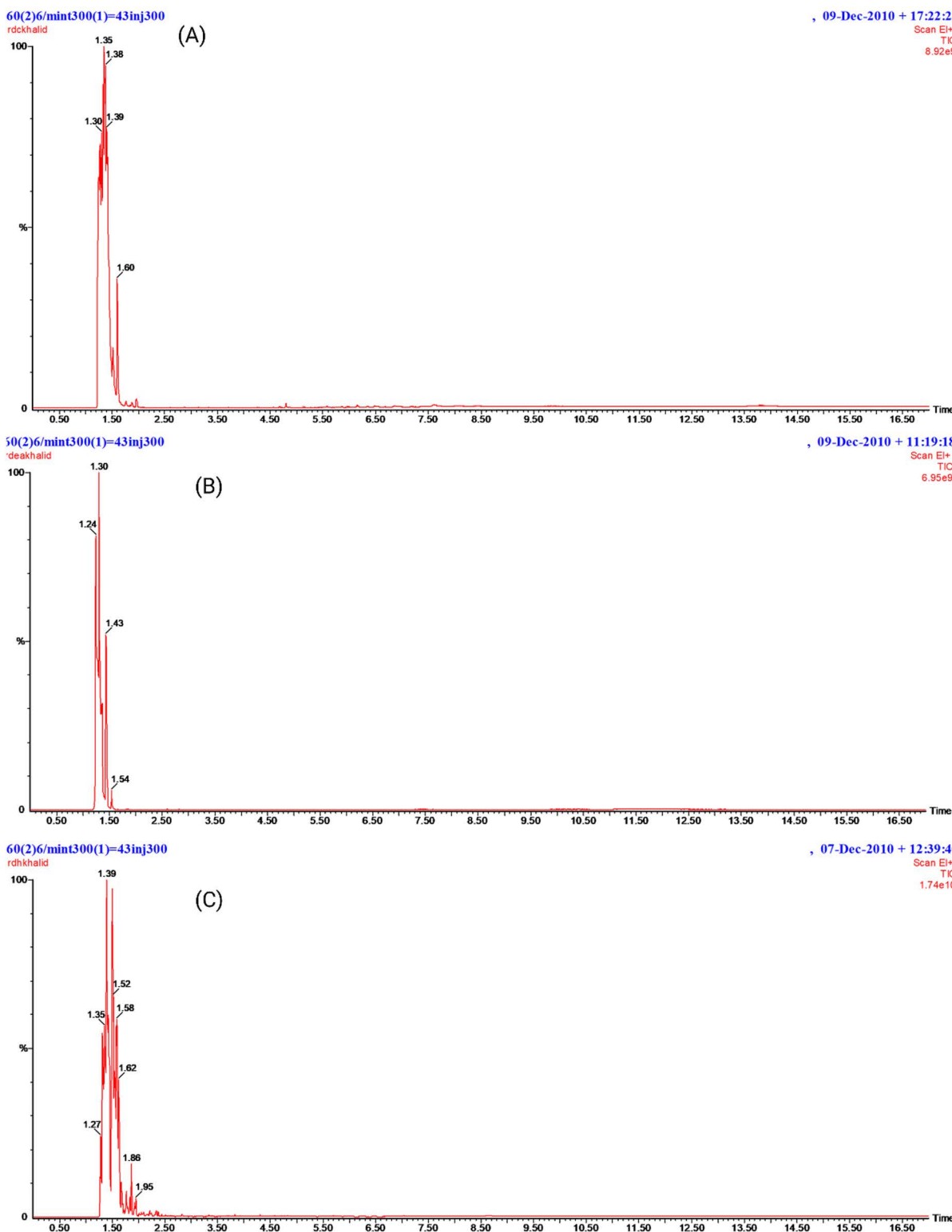

**Fig 2. GC-MS chromatogram of ethyl acetate, chloroform, and n-hexane extracts of *D. saeneb* (Forsk.) Hepper & J.R.I.Wood.** (A) Ethyl acetate extracts chromatogram, (B) Chloroform extracts chromatogram, (C) n-Hexane extracts chromatogram.

(A) Ethyl Acetate Extracts of *D. senab*

Heptyl hexacosyl ether

Hentriacontane

Carbonic acid, 2-ethylhexyl nonyl ester

Cyclohexyldimethoxymethyl- silane

Eicosyl heptyl ether

2-hexyl- 1-decanol

(B) Chlroform Extracts of *D. senab*

2,4-di-tert-butylphenol

Dotriacontane

Triacontane

Tetrapentacontane

Phytyl, 2-methylbutanoate

Eicosanoic acid

(C) n-Hexane Extracts of *D. senab*

Benzene, 1-ethyl-3-methyl-

Carbonic acid, decyl undecyl ester

Hentriacontane

**Fig 3. Names of compounds and their structures detected in ethyl acetate, chloroform, and n-hexane extracts of *D. saeneb* (Forsk.) Hepper & J.R.I.Wood.** All structures were created using ChemDraw software.

**Table 1. List of GC-MS detected compounds, their retention time (RT), area (%), retention index (RI), molecular weight (M.W) and formula in chloroform and n-hexane extracts of *D. saeneb* (Forsk.) Hepper & J.R.I.Wood.**

| Plant | Extracts | RT | Group | Compound | Area (%) | RI | M.W | Formula |
|---|---|---|---|---|---|---|---|---|
| *D. saeneb* | **Ethyl-Acetate** | 2.164 | Ethers | Heptyl hexacosyl ether | 5.0970 | 3358.00 | 480 | C33H68O |
| | | 2.324 | Alkanes | Hentriacontane | 8.8499 | 478.65 | 436 | C31H64 C18H36O3 |
| | | 2.594 | Carbonate esters | Carbonic acid, 2-ethylhexyl nonyl ester | 17.815 | 1953.00 | 300 | C31H64 |
| | | 2.825 | Alkanes | Hentriacontane | 6.8932 | 478.65 | 436 | C9H20O2Si |
| | | 3.145 | Silanes | Silane, cyclohexyldimethoxymethyl- | 24.497 | NIL | 188 | C31H64 |
| | | 3.36 | Alkanes | Hentriacontane | 1.4271 | 478.65 | 436 | C27H56O |
| | | 3.42 | Ethers | Eicosyl heptyl ether | 1.7092 | 2763.00 | 396 | C31H64 |
| | | 3.69 | Alkanes | Hentriacontane | 2.6133 | 478.65 478.65 | 436 | C31H64 |
| | | 3.76 | Alkanes | Hentriacontane | 4.8028 | 1504.00 | 436 | C16H34O |
| | | 4.43 | Alcohols | 1-decanol, 2-hexyl- | | | 242 | |
| | **Chloroform** | 4.8 | Phenol | 2,4-di-tert-butylphenol | 5.0208 | 1519. | 206 | C14H22O |
| | | 5.156 | Alkanes | Dotriacontane | 1.2222 | 489.99 | 450 | C32H88 |
| | | 5.576 | Alkanes | Triacontane | 2.3046 | 1488.4 | 422 | C30H62 |
| | | 6.156 | Esters | Phytyl, 2-methylbutanoate | 6.2362 | 2440.9 | 380 | C25H48O2 |
| | | 6.866 | Fatty acid | Eicosanoic acid | 4.0481 | 2380. | 312 | C20H40O2 |
| | | 7.612 | Alcohols | Phytol | 5.7720 | 2128. | 296 | C20H40O |
| | | 6.481 | Alkanes | Tetrapentacontane | 1.5249 | 489.99 | 450 | C32H66 |
| | **n-Hexane** | 2.209 | Aromatic | Benzene, 1-ethyl-3-methyl- | 22.467 | 963.9 | 120 | C9H12 |
| | | 2.334 | Carbonate esters | Carbonic acid, decyl undecyl ester | 11.839 | 2428 | 356 | C22H44O3 |
| | | 2.825 | Carbonate esters | Carbonic acid, decyl undecyl ester | 11.839 | 2428 | 356 | C22H44O3 |
| | | 3.335 | Carbonate esters | Carbonic acid, decyl undecyl ester | 11.839 | 2428 | 356 | C22H44O3 |
| | | 3.83 | Alkanes | Hentriacontane | 22.827 | 472.7 | 436 | C31H64 |
| | | 4.765 | Alkanes | Hentriacontane | 22.827 | 472.7 | 436 | C31H64 |
| | | 5.701 | Alkanes | Hentriacontane | 22.827 | 472.7. | 436 | C31H64 |

## Qualitative tests demonstrate that the leaves of *D. saeneb* possess different phytochemicals

The phytochemicals in various extracts (methanol, ethyl Acetate, and n-hexane) of *D. saeneb* (Forsk.) Hepper & J.R.I.Wood leaves were qualitatively assessed. The results indicate the presence of leucoanthocyanins, quinones, and lignins in all three extracts. Carbohydrates and terpenoids were identified in both the methanol and n-hexane extracts, while anthraquinone were found in both Methanol and n-Hexane extracts. while Phlobatannins were detected exclusively in the Ethyl acetate extracts, as shown in Table 2. Based on the GC-MS and qualitative results, we hypothesize that there may be a correlation between the plant extracts and total phenolic content, total tannins content, and total flavonoid content.

## Quantitative analysis of total phenols, total tannins, total flavonoids, and reducing sugar content in *D. saeneb* (Forsk.) Hepper & J.R.I.Wood leaf extracts

Medicinal plants contain a diverse array of phytochemicals, which confer various pharmacological activities, including anticancer, anti-inflammatory, antioxidant, and anti-diabetic effects. The quantitative measurement of these phytochemicals can be achieved through various assays. To explore the phytochemical composition of *D. saeneb* (Forsk.) Hepper & J.R.I.Wood leaves, researchers assessed the Total Flavonoid Content (TFC), Total Tannin Content (TTC), and Total Phenolic Content (TPC) in three different extracts: methanol, ethyl acetate, and n-hexane. The Folin-Ciocalteu reagent was used for these determinations.

Total flavonoid content (TFC) was expressed in terms of quercetin equivalents (mg of GAE/g of extracts). The standard curve equation for TFC in the standard was ($y = 0.0022x + 0.1409$) with ($R^2 = 0.9943$). For methanol extracts, the equation was ($y = 0.0007x + 0.0337$) with ($R^2 = 0.9946$). For ethyl acetate, it was ($y = 0.0005x + 0.1606$) with ($R^2 = 0.9937$). For n-hexane, the equation was ($y = 0.0004x + 0.0114$) with ($R^2 = 0.9936$). The results showed a

**Table 2. Qualitative Analysis of Methanolic, Ethyl acetate and n-Hexane Leaves Extract of *D. saeneb* (Forsk.) Hepper & J.R.I.Wood.**

| Constituents | Observations | Methanol | Ethyl Acetate | n-Hexane |
|---|---|---|---|---|
| Quinones | Greenish blue color | + | + | + |
| Glycosides | Reddish brown color | + | + | _ |
| Carbohydrates | Brick red precipitate | + | − | _ |
| Leucoanthocyanins | Upper red color | + | + | _ |
| Lignins | Yellow color | + | + | + |
| Emodins | Red color in ammoniacal layer | − | − | + |
| Anthraquinones | Pink, violet, or red color in ammoniacal layer | − | + | + |
| Amino Acids | Purple color | − | − | − |
| Phlobatannins | Red precipitates | − | + | − |
| Lactones | Deep red color | − | − | − |
| Anthocyanins | Pink red, then blue violet | − | − | − |
| Terpenoids | Reddish brown color at the interface | + | − | + |

+: Presence; -: Absence.

positive correlation between TFC in the extracts and increasing concentration. TFC in the standard ranged from 0.34 to 1.189 µg/mL. In methanol extracts, it ranged from 0.087 to 0.778 µg/mL. In ethyl acetate, it ranged from 0.188 to 0.624 µg/mL. In n-hexane, it ranged from 0.03 to 0.44 µg/mL. TTC in leaf extracts was expressed in terms of Gallic Acid equivalent (mg of GAE/g of extracts). The standard curve equation for TTC in the standard was ($y = 0.0052x + 0.2263$) with ($R^2 = 0.9979$). For methanol, the equation was ($y = 3E{-}05x + 0.0154$) with ($R^2 = 0.9832$). For ethyl acetate, it was ($y = 0.0001x + 0.0071$) with ($R^2 = 0.9883$). For n-Hexane, the equation was ($y = 0.0001x - 0.0067$) with ($R^2 = 0.9941$). These results also show a positive correlation of TTC in leaves extracts with increasing concentration. Additionally, the TTC in standard ranged from 0.462 to 2.359 µg/mL, in methanol extracts (0.016–0.048 µg/mL), in ethyl acetate (0.0196–0.153 µg/mL) and in n-Hexane (0.0012–0.134 µg/mL) as shown in Fig 4 E–H. TPC was expressed as mg/g Gallic acid equivalent using the standard curve equation with Gallic acid as the standard compound. The standard curve equation for TPC in the standard was $y = 0.0015x + 0.0806$, $R^2 = 0.9992$; for methanol: $y = 0.0002x + 0.0237$, $R^2 = 0.9914$; for Ethyl acetate: $y = 0.0005x - 0.0309$, $R^2 = 0.9909$; and for n-Hexane: $y = 0.0004x - 0.0372$, $R^2 = 0.9929$. TPC showed a positive correlation with increasing concentration among all extracts. Furthermore, the TPC in standard ranged from 0.11 to 0.38 µg/mL, in methanol extracts (0.042–0.267 µg/mL), in ethyl acetate (0.0275–0.487 µg/mL) and in n-Hexane (0.001–0.348 µg/mL) as depicted in Fig 4 IL.

The Reducing Sugar Content (RSC) in three different extracts (ethanol, methanol, and aqueous) of *D. saeneb* (Forsk.) Hepper & J.R.I.Wood and standard sucrose was expressed in terms of D-glucose equivalent (mg of GE/g of extract). A strong linear relationship was observed between the standard and the plant extracts with respect to RSC; however, the RSC was very low in the plant extracts compared to the standard. Specifically, the RSC, in ethanol extracts ranged from 0.005-0.05 ug/mL, in methanol (0.01-0.09 µg/mL) and in aqueous (0.005-0.035 µg/mL) while the RSC in the standard RSC ranged from 0.5-4.5 µg/mL), as depicted in Fig 5 A–D.

### Antioxidant effects of *D. saeneb* (Forsk.) Hepper & J.R.I.Wood leaf crude extracts through ABTS assay (*in vitro*)

The radical scavenging activity of ethanol and aqueous extracts of *D. saeneb* (Forsk.) Hepper & J.R.I.Wood leaves were evaluated using the ABTS assay. Both extracts inhibited ABTS in a dose-dependent manner, with the highest percentage of inhibition recorded for the aqueous extracts (83.254±4.021%) at a concentration of 250 µg/mL. In comparison to the ethanol extract, the extract demonstrated stronger ABTS inhibitory effects. Furthermore, at the same concentration (250 µg/mL), both extracts

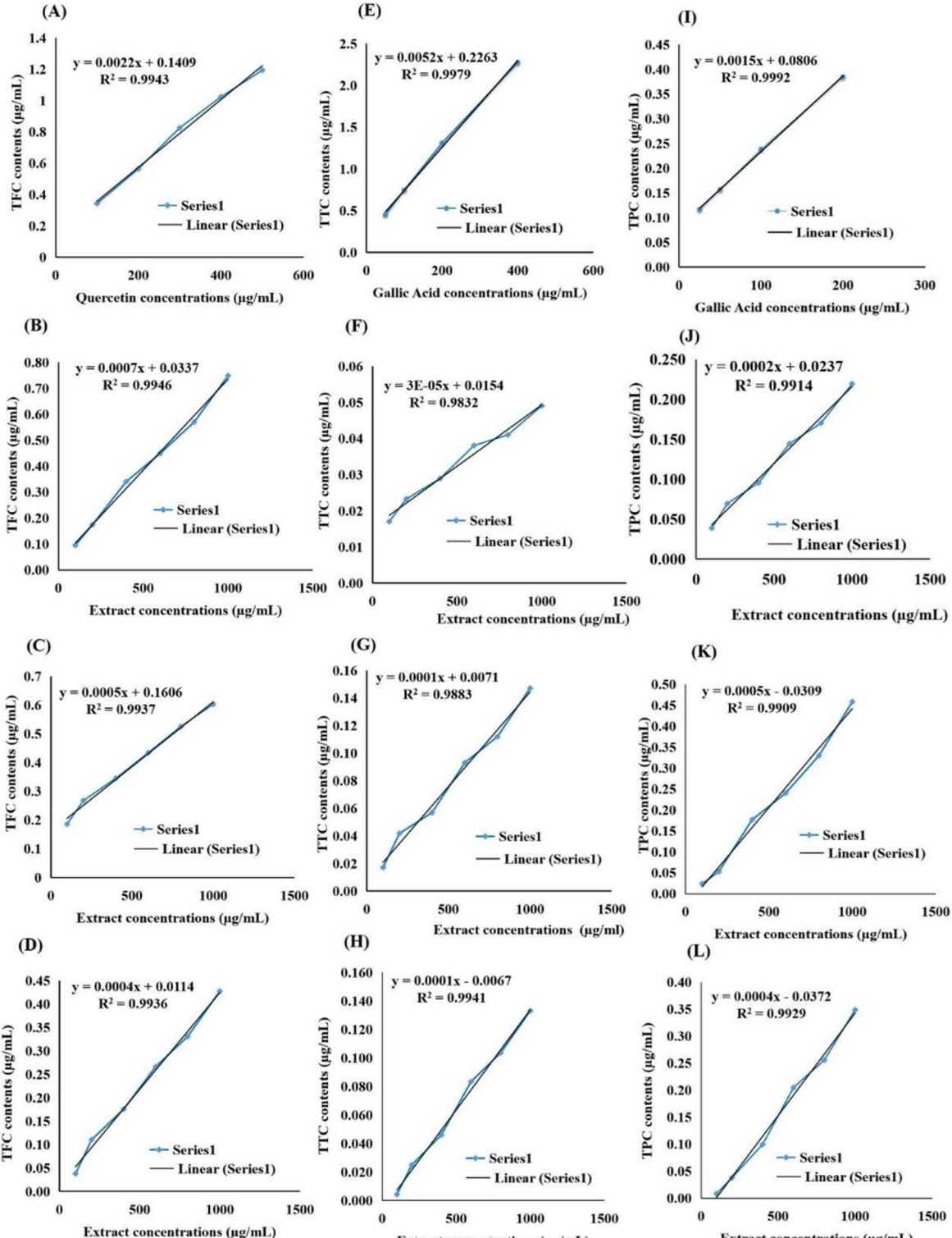

**Fig 4. Illustrates the correlation between various concentrations of *D. saeneb* (Forsk.) Hepper & J.R.I.Wood leaf extracts and Total Flavonoid Content (TFC), Total Tannin Content (TTC), and Total Phenolic Content (TPC).** (A) Presents the Quercetin standard correlation curve for TFC. (B), (C), and (D) show the correlation curves of TFC in methanol, ethyl acetate, and n-hexane extracts, respectively. (E) Represents the Gallic Acid standard

correlation curve for TTC. (F), (G), and (H) illustrate the correlation curves of TTC in methanol, ethyl acetate, and n-hexane extracts, respectively. (I) Represents the Gallic Acid standard correlation curve for TPC. (J), (K), and (L) display the correlation curves of TPC in methanol, ethyl acetate, and n-hexane extracts, respectively.

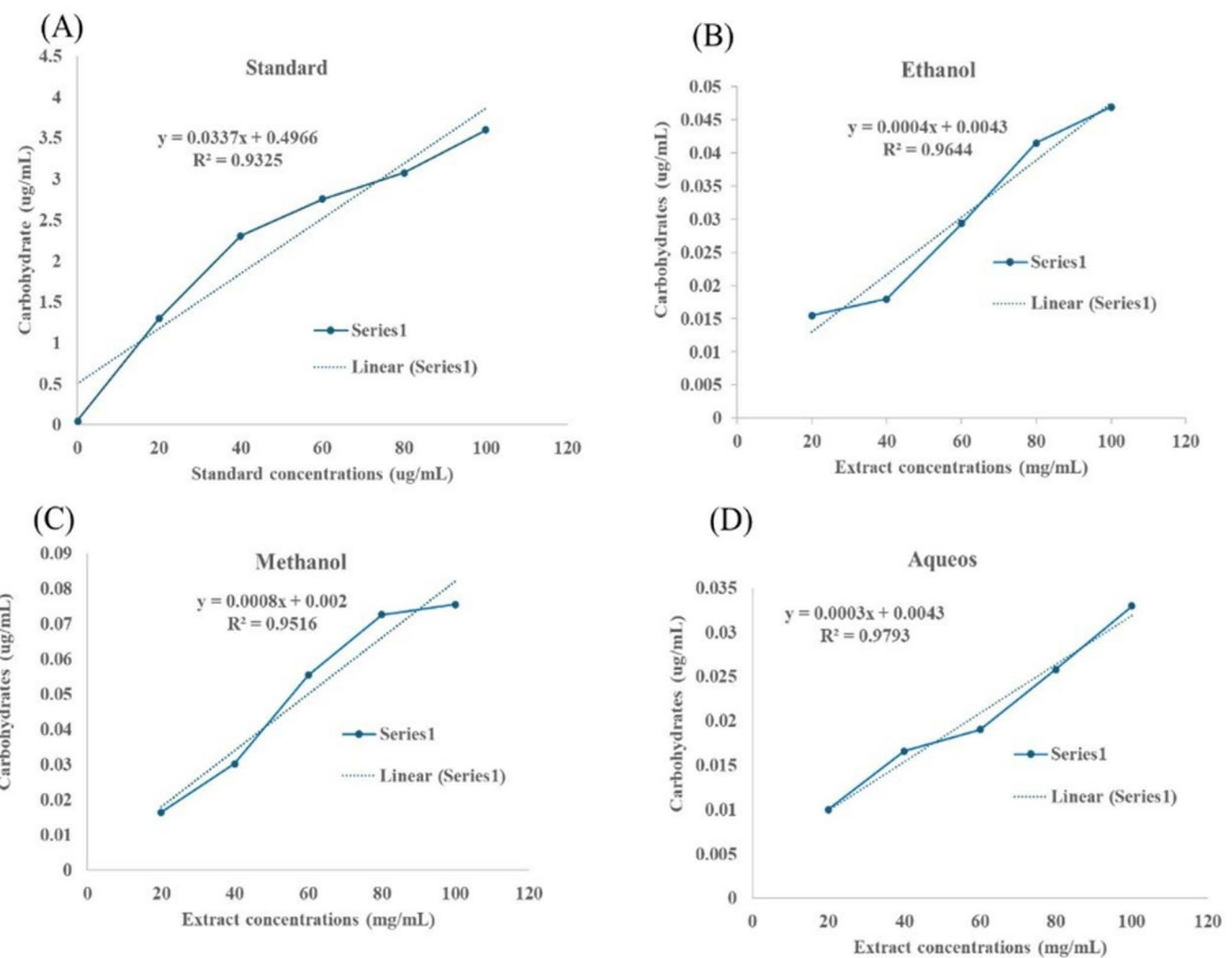

**Fig 5. Correlation between different concentrations of Standard/*D. saeneb* (Forsk.) Hepper & J.R.I.Wood leaf extracts with RSC.** (A) RSC in standard sucrose, (B) RSC in ethanol extract, (C) RSC in methanol, and (D) RSC in Aqueous extracts.

exhibited significantly stronger effects compared to the standard (ascorbic acid). The estimated IC50 values were 91.455 µg/mL for the aqueous extract and 98.3949 µg/mL for the ethanol extract. These results are further summarized in Fig 6.

### Antioxidant effect of *D. saeneb* (Forsk.) Hepper & J.R.I.Wood leaf extracts: DPPH (*in vitro*) inhibition

Medicinal plants contain various phytochemicals with radical scavenging activities. The extracts of *D. saeneb* (Forsk.) Hepper & J.R.I.Wood (Methanol, Ethyl Acetate, and n-Hexane) possess distinct phytochemicals, and their radical

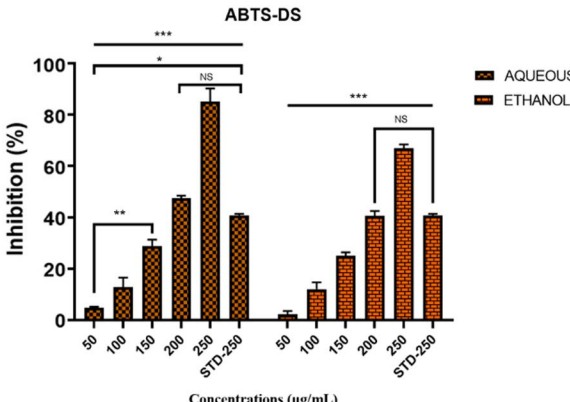

**Fig 6. Percent (%) inhibition of ABTS by aqueous and ethanol extracts of *D. saeneb* (Forsk.) Hepper & J.R.I.Wood leaves at various concentrations (µg/mL) compared to standard ascorbic acid.** The experiments were conducted three times, and the results are presented as mean ± standard deviation (SD). Statistical significance is denoted as follows: NS (not significant), *p < 0.05, **p < 0.001, ***p < 0.0001.

scavenging activities were assessed using the DPPH assay. The results demonstrate that all extracts inhibit DPPH in a dose-dependent manner, with no significant variation among the groups for the methanol and n-hexane extracts. However, a significant increase in inhibition was observed at a high dose (500 µg/mL) for the ethyl acetate extract. Furthermore, the highest level of DPPH inhibition was observed in the ethyl acetate extract, while the inhibition levels in the methanol and n-hexane extracts were lower in comparison to those of the ethyl acetate extract, as depicted in Fig 7A and Table 3.

### Antidiabetic effects of *D. saeneb* (Forsk.) Hepper & J.R.I.Wood leaf extracts through alpha amylase inhibition

Medicinal plants contain various active compounds with diverse pharmacological activities, including the inhibition of alpha-amylase, which reduces the absorption of glucose from the intestine into the bloodstream. In this study, different extracts of *D. saeneb* (Forsk.) Hepper & J.R.I.Wood leaves at leaves—specifically methanol, ethyl acetate, and n-hexane were evaluated for their alpha-amylase inhibitory effects at varying concentrations (125, 250, and 500 µg/mL), and the results were compared to the standard acarbose. The results indicate that the methanol extracts dose-dependently inhibit alpha-amylase levels, though insignificantly different among groups at various concentrations, and the inhibition is significantly lower than that observed with standard inhibitor acarbose. In contrast, the ethyl acetate extracts demonstrate a significant and dose-dependent inhibition of alpha-amylase activity across various concentrations. Notably, at the highest concentration, the level of inhibition observed is significantly greater (approximately 14% difference) than that produced by the standard acarbose. Similarly, the n-hexane extracts also exhibit a significant and dose-dependent inhibition of alpha-amylase levels when compared to one another, with a considerable difference (nearly 40%) between the high concentration and the standard. In conclusion, the ethyl acetate extract exhibits the most pronounced inhibition at elevated concentrations, while the effects of methanol and n-hexane extracts are relatively comparable regarding alpha-amylase inhibition, as depicted in Fig 7B and Table 4.

### Discussions

Medicinal flora are key providers of bioactive compounds that possess therapeutic properties, rendering them indispensable for both ancestral and contemporary healing practices [26,27]. Their importance is attributed to their effectiveness and affordability [28]. The composition of volatile compounds in medicinal plants varies with seasonal changes

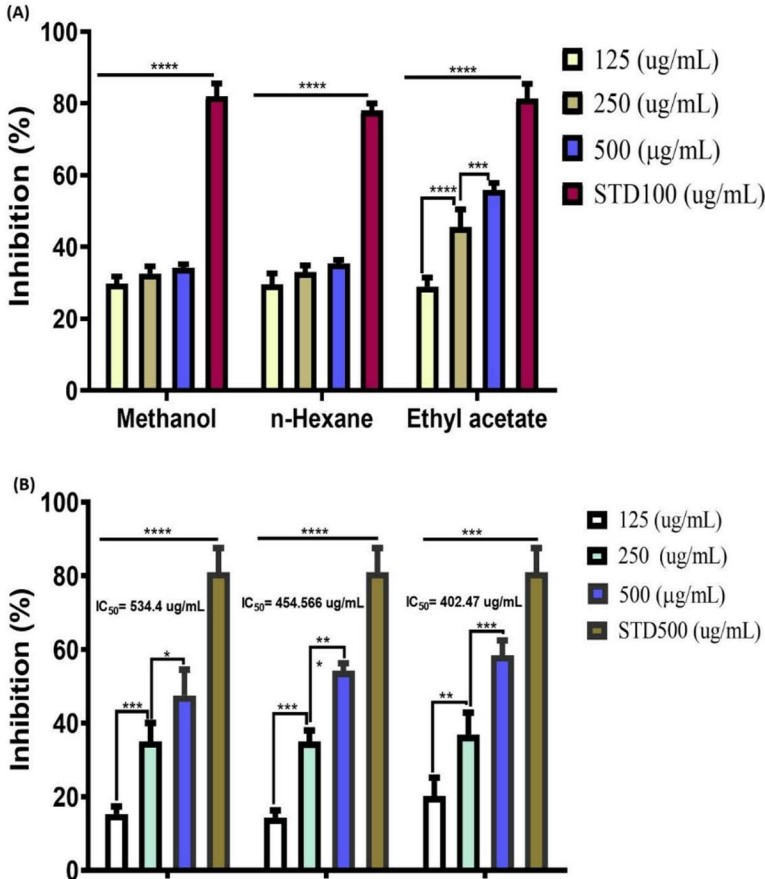

**Fig 7. The inhibitory effects of *D. saeneb* (Forsk.) Hepper & J.R.I.Wood leaf extracts (methanol, ethyl acetate and n-hexane) on (A). DPPH and (B). alpha amylase enzyme activity.** The combined results of three independent experiments are expressed as mean ± standard deviation (SD). Statistical significance was determined at $p < 0.05$ (*$p < 0.05$, ***$p < 0.001$, and NS = non-significant).

**Table 3. IC50 values of *In vitro* DPPH activity of *D. seaneb* (Forsk.) Hepper & J.R.I.Wood leaves extracts.**

| Plant species | Extracts | Concentration (µg/ml) | Inhibition (%) | IC50 (µg/ml) |
|---|---|---|---|---|
| *Debregeasia seaneb* (Forsk.) Hepper & J.R.I.Wood | Methanol | 125 | 15.32 ± 0.01 | 534.4 |
| | | 250 | 35.01 ± 0.02 | |
| | | 500 | 47.53 ± 0.01 | |
| | n-Hexane | 125 | 14.31 ± 0.03 | 454.566 |
| | | 259 | 31.99 ± 0.04 | |
| | | 500 | 54.21 ± 0.01 | |
| | Ethyl acetate | 125 | 20.21 ± 0.02 | 402.47 |
| | | 250 | 36.79 ± 0.03 | |
| | | 500 | 58.45 ± 0.02 | |

**Table 4. IC50 values of alpha amylase activity of *D. saeneb* (Forsk.) Hepper & J.R.I.Wood leaves extracts.**

| Plant species | Extracts | Concentration (µg/ml) | Inhibition (%) | IC50 (µg/ml) |
|---|---|---|---|---|
| *Debregeasia saeneb* (Forsk.) Hepper & J.R.I.Wood | Methanol | 125 | 29.79 ± 0.02 | 1925.68 |
| | | 250 | 32.62 ± 0.01 | |
| | | 500 | 34.18 ± 0.02 | |
| | n-Hexane | 125 | 29.59 ± 0.04 | 1472.78 |
| | | 250 | 32.93 ± 0.06 | |
| | | 500 | 35.41 ± 0.02 | |
| | Ethyl acetate | 125 | 29.27 ± 0.04 | 388.305 |
| | | 250 | 45.53 ± 0.06 | |
| | | 500 | 55.87 ± 0.02 | |

and different stages of plant development. These compounds play a crucial role in deterring pest, suppress the growth of nearby flora, and lure pollinators [29]. Researchers have developed a range of methodologies to identify and extract these bio-active compounds from plants [25]. Such methodologies include techniques such as chromatographic analysis, spectral analysis, and various extraction methods, which enable the precise identification and procurement of compounds of pharmacological relevance [25].

GC-MS is a widely utilized technique for detecting compounds in medicinal plant extracts. In the current study, GC-MS was employed to identify the phytochemicals present in the leaf extracts of *D. saeneb* (Forsk.) Hepper & J.R.I.Wood, an area that has not been previously explored. The results indicate the detection of six compounds in the ethyl acetate extract, seven compounds in the chloroform extract, and three compounds in the n-hexane extract. Notably, hentriacontane was detected in both the ethyl acetate and n-hexane extracts at different retention times, suggesting that this compound may be present in high concentrations in the leaf extracts of *D. saeneb* (Forsk.) Hepper & J.R.I.Wood. As the GC-MS results indicate the presence of various phytochemicals in different extracts, we aimed to confirm some of these compounds qualitatively by performing different tests. The extracts of *D. saeneb* (Forsk.) Hepper & J.R.I.Wood were found to contain a variety of phytochemicals, as listed in Table 2, while others have been reported previously from the same plant [15,16,30].

Additionally, the results of the current study demonstrated that tannins, phenolics, and flavonoids were present in varying concentrations in different extracts of *D. saeneb* (Forsk.) Hepper & J.R.I.Wood leaves. These compounds are commonly found in various plants and are recognized for their diverse health benefits [31]. The natural compounds, including phenolics, flavonoids, and tannins, exhibit a range of pharmacological activities, such as anti-diabetic, anti-allergic, anti-bacterial, anticancer, and anti-inflammatory properties [32,33]. A plethora of research reveals that plant phytochemicals reduce blood glucose levels through different mechanisms, including the restoration of β-cells and liver cells, inhibition of α-amylase and β-glucosidase, and prevention of oxidation and hyperlipidemia [9,34]. Following the evaluation of compounds in different extracts, it was important to assess the pharmacological activities of these extracts. In these activities, alpha-amylase inhibition was checked for its anti-diabetic effect, and DPPH as well as ABTS levels were assessed for antioxidant effects. The literature revealed that *D. saeneb* (Forsk.) Hepper & J.R.I.Wood has not been studied for its alpha-amylase inhibitory effect and antioxidant properties through DPPH in these extracts. Furthermore, there are numerous plant extracts known to have healing effects on diabetes through alpha-amylase inhibition, DPPH and ABTS [34–36]. According to the DPPH results, all extracts demonstrated dose-dependent antioxidant activities, with the methanol extract exhibiting the highest antioxidant activity among them. This aligns with previous reports indicating that methanol extracts often possess high antioxidant effects [37–39], possibly due to their inherent nature [40]. Furthermore, alpha-amylase plays a crucial role in the digestion of carbohydrate polymers in the gastrointestinal tract, facilitating their absorption into the bloodstream and subsequent elevation of blood glucose levels. Therefore, the inhibition of alpha-amylase serves as a primary therapeutic target for reducing

carbohydrate digestion. By decreasing carbohydrate digestion, blood glucose levels can be effectively lowered [23]. Various compounds found in plants contribute to the reduction of blood glucose levels through diverse mechanisms [9]. Phenolic compounds, including phenolic acids and flavonoids, can bind to alpha-amylase and alter its activity [24]. The results of our study indicate that all extracts exhibit a dose-dependent inhibitory effect on alpha-amylase. This effect may be attributed to the presence of various phytochemicals within the extracts [41].

## Conclusions

In conclusion, the leaf extracts of *D. saeneb* (Forsk.) Hepper & J.R.I.Wood underwent a comprehensive analysis, revealing the presence of 14 distinct phytochemicals through GC-MS. These extracts exhibited substantial levels of Total Phenolic Content (TPC), Total Tannin Content (TTC), and Total Flavonoid Content (TFC), all of which are known for their potential pharmacological benefits. Notably, our findings suggest that these phytochemicals play a pivotal role in inhibiting alpha-amylase and DPPH, which may contribute to reducing blood glucose levels and protecting the pancreas from damage. This study marks the first detection of these compounds using a combination of GC-MS, qualitative, and quantitative tests, further underscoring their significance. We strongly recommend that future research delves deeper into the pharmacological activities of the compounds present in *D. saeneb* (Forsk.) Hepper & J.R.I.Wood, particularly in areas such as inflammation, cancer, and various molecular mechanisms. This exploration could pave the way for promising treatments, especially in diabetes management, pending further investigations and validation.

## Acknowledgments

The authors extend their sincere appreciation to the Deanship of Scientific Research at King Saud University for funding this work through the Vice Deanship of Scientific Research Chairs, Chair of Biomedical Applications of Nanomaterials. We are also grateful to COMSATS University's GC-MS Laboratory for granting our students access to their paid GC-MS services. Special thanks to Dr. Hossam Ebaid for his valuable assistance with the statistical analysis.

## Author contributions

**Conceptualization:** Bashir Ahmad, Rashid Khan, Rabia Afza, Mostafa A Abdel-Maksoud.

**Data curation:** Rashid Khan, Sumaira Miskeen, Khalid Ahmad.

**Formal analysis:** Bashir Ahmad, Rashid Khan, Khalid Ahmad, Mohamed Y. Zaky.

**Funding acquisition:** Bashir Ahmad, Mostafa A Abdel-Maksoud, Salman Alrokayan.

**Investigation:** Rashid Khan, Sumaira Miskeen, Khalid Ahmad.

**Methodology:** Rashid Khan, Sumaira Miskeen, Khalid Ahmad.

**Project administration:** Bashir Ahmad, Mostafa A Abdel-Maksoud, Salman Alrokayan.

**Resources:** Bashir Ahmad, Mostafa A Abdel-Maksoud.

**Software:** Bashir Ahmad, Rabia Afza, Mohamed Y. Zaky, Salman Alrokayan.

**Supervision:** Bashir Ahmad, Rabia Afza, Salman Alrokayan.

**Validation:** Bashir Ahmad, Rabia Afza, Sumaira Miskeen, Mostafa A Abdel-Maksoud.

**Visualization:** Bashir Ahmad, Rashid Khan, Mohamed Y. Zaky.

**Writing – original draft:** Rashid Khan, Mohamed Y. Zaky.

**Writing – review & editing:** Bashir Ahmad, Rabia Afza, Mostafa A Abdel-Maksoud, Mohamed Y. Zaky, Salman Alrokayan.

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
