## [Decision Letter · Decision Letter 0]

17 Dec 2024

PONE-D-24-51482Phytochemical Composition and Bioactivity of Debregeasia saenab  Leaves: Insights into Anti-Diabetic and Antioxidant PropertiesPLOS ONE

Dear Dr. Ahmad,

Thank you for submitting your manuscript to PLOS ONE. After careful consideration, we feel that it has merit but does not fully meet PLOS ONE’s publication criteria as it currently stands. Therefore, we invite you to submit a revised version of the manuscript that addresses the points raised during the review process.

We look forward to receiving your revised manuscript.

Kind regards,

Lakshmanan Govindan

Academic Editor

PLOS ONE

**Journal Requirements:**

We are very thankful to Higher Education of Pakistan for financial support (grant reference number: 533/IPFP-II(Batch-I)/NAHE/HEC/2020/126). The authors extend their appreciation to the Researchers Supporting Project number (RSPD2023R725) King Saud University, Riyadh, Saud Arabia.

We are very thankful to Higher Education of Pakistan for financial support (grant reference number: 533/IPFP-II(Batch-I)/NAHE/HEC/2020/126). The authors extend their appreciation to the Researchers Supporting Project number (RSPD2023R725) King Saud University, Riyadh, Saud Arabia.

We are very thankful to Higher Education of Pakistan for financial support (grant reference number: 533/IPFP-II(Batch-I)/NAHE/HEC/2020/126). The authors extend their appreciation to the Researchers Supporting Project number (RSPD2023R725) King Saud University, Riyadh, Saud Arabia.

6. We noted in your submission details that a portion of your manuscript may have been presented or published elsewhere. "7117" Please clarify whether this [conference proceeding or publication] was peer-reviewed and formally published. If this work was previously peer-reviewed and published, in the cover letter please provide the reason that this work does not constitute dual publication and should be included in the current manuscript.

Reviewers' comments:

Reviewer's Responses to Questions

**Comments to the Author**

1. Is the manuscript technically sound, and do the data support the conclusions?

Reviewer #1: Yes

Reviewer #2: Partly

Reviewer #3: No

Reviewer #4: No

2. Has the statistical analysis been performed appropriately and rigorously? 

Reviewer #1: Yes

Reviewer #2: No

Reviewer #3: No

Reviewer #4: Yes

3. Have the authors made all data underlying the findings in their manuscript fully available?

Reviewer #1: Yes

Reviewer #2: Yes

Reviewer #3: No

Reviewer #4: Yes

4. Is the manuscript presented in an intelligible fashion and written in standard English?

Reviewer #1: Yes

Reviewer #2: No

Reviewer #3: No

Reviewer #4: No

5. Review Comments to the Author

**Reviewer #1:**  This study provides a comprehensive analysis of the phytochemical composition and biological activities of Debregeasia saenab, highlighting its antioxidant and antidiabetic potential for the first time. Using Gas Chromatography-Mass Spectrometry (GC-MS), bioactive compounds such as glycosides, terpenoids, and flavonoids were identified in extracts obtained with different solvents, with the methanol extract demonstrating the highest antioxidant activity. These findings scientifically validate the plant's traditional uses and offer a valuable foundation for drug development. However, the study has several limitations. The findings were assessed only at the in vitro level, without in vivo validation. Moreover, the molecular mechanisms underlying the antioxidant and antidiabetic activities were not explored in detail, and the influence of different solvents on biological activity was not sufficiently discussed. Additionally, the lack of safety and toxicity assessments leaves a significant gap regarding therapeutic applications. Another limitation is the absence of an evaluation of the reproducibility of the study's findings across different laboratories.

In addition to the positive aspects of the study, the following deficiencies must also be addressed. The species name in the title and throughout the text is incorrectly provided as "Debregeasia saenab," whereas the correct species epithet should be "saeneb." This represents a significant scientific error within the article. Furthermore, in the Materials and Methods section, the collection sites of the plant must be presented in accordance with scientific standards. The collection habitat, coordinates, altitude, and collector number should be explicitly stated. Additionally, information regarding the drying process of the collected leaves is missing. It must be clarified whether the leaves were air-dried and, if so, whether the drying was conducted in a shaded environment, and in what specific setting (e.g., drying room, drying rack).

**Reviewer #2:**  1. Authors need to give the proper rationale for the selection of current research drug and the study. Just not reported to the date can't be the rationale for the study.

2. Authors need to produce the collection site GPS coordinates and details of voucher specimen for future reference.

3. Authors need to mention the proper rationale for the selection of extraction solvents and method of extraction (maceration).

4. Authors also need to give the details of order of solvents used for the extraction.

5. Mention the reason for moistening of filter paper with sodium phosphate before filter paper.

6. Authors are suggested to check the chemical names/formulas and botanical names for typos.

7. Table 2: Can glycosides and carbohydrates be detected in n-hexane extract?

8. Justify the presence of carbohydrates and terpenoids in n-hexane and methanol extract but not in ethyl acetate extract during sequential extraction.

9. Table 3 and 4- Check the IC50 values of test extracts with reference to the percent inhibition. E.g. Table3- Methanolic extract gives 47.53 % inhibition at 500 microgrm/ml but IC50 value mentioned is 190.95. Justify.

10. Authors are recommended to discuss the in-vitro assays results with reference to the recent literature showing similar studies. Authors are suggested to read the following articles for better clarity;

https://doi.org/10.1016/j.jics.2022.100807

https://doi.org/10.1016/j.sajb.2023.01.005

https://doi.org/10.1016/j.jep.2024.118055

https://doi.org/10.3390/ph16121636

https://doi.org/10.1016/j.prmcm.2023.100286

11. Authors need to revise the conclusion with reference to the crucial findings and future prospects of the study.

**Reviewer #3:**  The idea of screening plants for bioactive compounds with potential applications in food additives and medicine is both ancient and valuable. In this paper, the authors claim to explore the antioxidative and amylase inhibitory properties of leaves from Debregieasia saenab, suggesting potential pharmaceutical applications to mitigate diabetes-related effects. This claim is prominently displayed in the graphical abstract, yet it is not substantiated by the experiments presented in the manuscript. While some in vitro assays are included, they do not provide sufficient evidence to support the proposed pharmaceutical use of the plant.

Key Concerns and Suggestions:

1. Unvalidated Pharmaceutical Claim:

- The manuscript’s graphical abstract implies a pharmaceutical application for the leaves of Debregieasia saenab, yet no experiments have been conducted to test this claim. The study would benefit greatly from focused experiments that directly test bioactivities relevant to pharmaceutical use, such as cell-based or animal model studies. Without these, the claim is speculative and misleading.

2. In Vitro Experiments:

- The authors include some in vitro assays for antioxidative effects and alpha-amylase inhibition. However:

- The methodologies for these assays lack adequate detail and validation. For example, standard controls and benchmarks are not clearly outlined.

- The relevance of the in vitro results to real-world pharmaceutical applications is not discussed, leaving the interpretation of these findings vague and disconnected from the proposed claim.

3. Methodological Rigor:

- The primary method for compound identification—GC/MS—lacks clear validation. There is no evidence of standard use for peak identification, such as co-injection with known standards or database matching with validated spectra.

- The rationale for the selection of extraction methods is unclear, and the absence of overlap in identified compounds among the three extraction methods raises concerns about their reliability. A detailed comparison of the extraction methods’ efficiency and appropriateness is needed.

4. Colorimetric Tests and Standard Curves:

- The manuscript includes colorimetric tests to estimate the presence of compound groups (e.g., phenolics). However, these tests are not supported by proper validation, and the choice of standard curves seems arbitrary and unrelated to the compounds under study. The authors should clearly justify the assays and relate them to the proposed bioactivities.

5. Sampling and Representation:

- The collection of plant material is inadequately described, with no details about the number of plants sampled, their geographical location, or the sample size. This makes it impossible to determine whether the findings are representative of a single plant or a broader population. Without this information, the study’s conclusions about the pharmaceutical potential of the plant remain tenuous.

6. Language and Presentation:

- The manuscript suffers from imprecise language and a lack of adherence to scientific writing conventions. A thorough revision for clarity, grammar, and technical accuracy is needed.

- The structure and presentation fall short of the standards required for a journal like PLOS. Familiarity with similar high-quality publications in this field would be beneficial for aligning the manuscript with accepted norms.

Summary:

The paper addresses an important topic but suffers from significant methodological and reporting shortcomings. To make this work acceptable for publication:

• Provide direct experimental evidence to support pharmaceutical claims, such as cell-based or in vivo assays.

• Use validated methods for compound identification and quantification, with proper controls.

• Clearly justify the selection of extraction methods and evaluate their effectiveness.

• Provide detailed information about plant material collection and sampling.

• Revise the manuscript thoroughly for clarity, rigor, and alignment with journal standards.

I encourage the authors to address these issues comprehensively and resubmit after significant revision. This work has potential, but it requires much greater rigor to substantiate its claims and ensure its scientific credibility.

**Reviewer #4: ** The manuscript " Phytochemical Composition and Bioactivity of Debregeasia saenab Leaves: Insights into Anti-Diabetic and Antioxidant Properties" has several deficiencies.

The manuscript is inadequately presented and lacks a coherent rationale. The authors claim to have conducted biological activity; however, no such activity has been executed. The authors noted that ethnobotanical surveys indicate that D. saenab is traditionally employed to treat several ailments, including dermatitis, carbuncles, boils, bone fractures, skin rashes, eczema, pimples, tumors, and urinary tract disorders. The authors failed to acknowledge that the plant is traditionally utilized for diabetes management. What methodology did the authors employ to assess enzyme inhibition activities? The writers have indicated. The authors state in the abstract that their study provides extensive insights into the phytochemical composition of D. saenab leaves and its prospective therapeutic applications, including anti-diabetic and antioxidant benefits; however, such data is not thoroughly reported. The authors did not provide any data regarding the secondary metabolites implicated in the antidiabetic effect. The introduction must be explicit and aligned with the study's objective. The investigation lacks a distinct purpose, resulting in an unclear introduction. The results and discussion sections evidently lack distinct points.

6. PLOS authors have the option to publish the peer review history of their article (what does this mean? ). If published, this will include your full peer review and any attached files.

**Do you want your identity to be public for this peer review?** For information about this choice, including consent withdrawal, please see our Privacy Policy .

Reviewer #1: **Yes: ** Selami Selvi

Reviewer #2: **Yes: ** Shailendra Gurav

Reviewer #3: No

Reviewer #4: No

---

## [Author Response · Author response to Decision Letter 1]

7 Feb 2025

Dear Editor

We sincerely appreciate the time and effort you and the reviewers have dedicated to evaluating our manuscript, “Phytochemical composition and bioactivity of Debregeasia saeneb leaves: Insights into anti-diabetic and antioxidant properties”. We are grateful for the constructive feedback and insightful suggestions, which have helped us improve the quality and clarity of our work. We have carefully addressed all the comments and revised the manuscript accordingly.

Journal Requirements:

Reply: We appreciate the Editor efforts. We have make changes according to https://journals.plos.org/plosone/s/file?id=ba62/PLOSOne_formatting_sample_title_authors_affiliations.pdf.

2.In your Methods section, please provide additional information regarding the permits you obtained for the work. Please ensure you have included the full name of the authority that approved the field site access and, if no permits were required, a brief statement explaining why.

Reply: No specific permits were required for this study as the field site is publicly accessible, and the plant material collected is not classified as endangered or protected under local, national, or international regulations. Additionally, all research activities complied with institutional and ethical guidelines for sustainable and responsible scientific practices.

Reply: Thank you for pointing this out. The study is supported by the Higher Education Commission of Pakistan (grant reference number: 533/IPFP-II(Batch-I)/NAHE/HEC/2020/126) and by the Researchers Supporting Project number (RSPD2023R725) at King Saud University, Riyadh, Saudi Arabia. We ensured that the correct grant numbers are provided in the ‘Funding Information’ section upon resubmission.

We are very thankful to Higher Education of Pakistan for financial support (grant reference number: 533/IPFP-II(Batch-I)/NAHE/HEC/2020/126). The authors extend their appreciation to the Researchers Supporting Project number (RSPD2023R725) King Saud University, Riyadh, Saud Arabia.

Reply: We are very thankful to the Editor for correction. We have changed the Acknowledgement and funding section in the revised manuscript.

We are very thankful to Higher Education of Pakistan for financial support (grant reference number: 533/IPFP-II(Batch-I)/NAHE/HEC/2020/126). The authors extend their appreciation to the Researchers Supporting Project number (RSPD2023R725) King Saud University, Riyadh, Saud Arabia.

We are very thankful to Higher Education of Pakistan for financial support (grant reference number: 533/IPFP-II(Batch-I)/NAHE/HEC/2020/126). The authors extend their appreciation to the Researchers Supporting Project number (RSPD2023R725) King Saud University, Riyadh, Saud Arabia.

Reply: We are very thankful to the Editor for correction. We have changed the Acknowledgement and funding section in the revised manuscript.

6.We noted in your submission details that a portion of your manuscript may have been presented or published elsewhere. "7117" Please clarify whether this [conference proceeding or publication] was peer-reviewed and formally published. If this work was previously peer-reviewed and published, in the cover letter please provide the reason that this work does not constitute dual publication and should be included in the current manuscript.

Reply: Thank you for your comment. We would like to clarify that a similar abstract was previously published in ACS Omega, focusing on a different plant. While some lines may overlap, the majority of the content in the current manuscript has not been published or presented elsewhere. This work is part of a PhD thesis, and we have been cautious about publishing it until the student completes their degree. We believe this manuscript does not constitute dual publication, as it presents new and significant content beyond the prior abstract.

Review Comments to the Author

Reviewer #1: This study provides a comprehensive analysis of the phytochemical composition and biological activities of Debregeasia saenab, highlighting its antioxidant and antidiabetic potential for the first time. Using Gas Chromatography-Mass Spectrometry (GC-MS), bioactive compounds such as glycosides, terpenoids, and flavonoids were identified in extracts obtained with different solvents, with the methanol extract demonstrating the highest antioxidant activity. These findings scientifically validate the plant's traditional uses and offer a valuable foundation for drug development. However, the study has several limitations.

1.The findings were assessed only at the in vitro level, without in vivo validation. Moreover, the molecular mechanisms underlying the antioxidant and antidiabetic activities were not explored in detail, and the influence of different solvents on biological activity was not sufficiently discussed.

Reply: Thank you for your thoughtful comment regarding the in vivo validation, molecular mechanisms, and solvent influence on biological activity. We agree that in vivo validation is critical, and while the current manuscript focuses on the identification and preliminary characterization of bioactive compounds through GC-MS and in vitro assays, we acknowledge the need for further validation. In vivo experiments have already been conducted as part of this research, and we are preparing a separate manuscript to present those findings. Additionally, we appreciate your point about the molecular mechanisms underlying the antioxidant and antidiabetic activities. While this study primarily aims to establish a foundational understanding of the bioactive compounds, we intend to delve deeper into these mechanisms in future work. Regarding the influence of different solvents, we will ensure to expand on this aspect in the follow-up research, as it is a crucial factor in understanding the biological activity of the compounds. We believe the current findings provide valuable insights and serve as a starting point for further investigations. Thank you for your understanding, and we look forward to addressing these aspects in more detail in future publications.

Additionally, the lack of safety and toxicity assessments leaves a significant gap regarding therapeutic applications. Another limitation is the absence of an evaluation of the reproducibility of the study's findings across different laboratories.

Reply: Thank you for your valuable comments. While we have not yet conducted toxicity assessments on the leaves, we have evaluated the toxicity of the stem bark extracts, which showed no major risks. However, we are continuing to work on this and will include a more thorough investigation of the leaves in future studies. Regarding the reproducibility of our findings across different laboratories, we agree that this is an important consideration. As this study serves as a foundational step, we hope that subsequent research in other labs will replicate and build upon our results, further validating the findings. We are committed to addressing these aspects in our future work. Thank you again for your insightful feedback.

2.In addition to the positive aspects of the study, the following deficiencies must also be addressed. The species name in the title and throughout the text is incorrectly provided as "Debregeasia saenab," whereas the correct species epithet should be "saeneb." This represents a significant scientific error within the article.

Reply: Thank you for pointing out this error. We sincerely apologize for the mistake and have corrected the species name from "saenab" to "saeneb" throughout the entire manuscript. We appreciate your attention to detail and will ensure such errors are avoided in future submissions.

3.Furthermore, in the Materials and Methods section, the collection sites of the plant must be presented in accordance with scientific standards. The collection habitat, coordinates, altitude, and collector number should be explicitly stated.

Reply: Thank you for your comment. We have revised the manuscript to present the collection sites of the plant in accordance with scientific standards. The collection habitat, coordinates, altitude, and collector number have all been explicitly included in the revised manuscript. We appreciate your suggestion and hope the updated information meets the required standards.

4.Additionally, information regarding the drying process of the collected leaves is missing. It must be clarified whether the leaves were air-dried and, if so, whether the drying was conducted in a shaded environment, and in what specific setting (e.g., drying room, drying rack).

Reply. Thank you for highlighting this. We have now clarified the drying process of the collected leaves in the revised manuscript. We hope this additional information addresses your concern.

Reviewer #2: 

1. Authors need to give the proper rationale for the selection of current research drug and the study. Just not reported to the date can't be the rationale for the study.

Reply: Thank you for your insightful comment. We appreciate the opportunity to address your concern. In the revised manuscript, we have clarified and expanded the rationale for selecting Debregeasia saeneb as the subject of this study. We have provided a clear justification based on its potential bioactive properties, traditional uses, and gaps in the current research, going beyond the novelty aspect to emphasize its relevance for therapeutic applications. This additional explanation has been included at the end of the Introduction section.

2. Authors need to produce the collection site GPS coordinates and details of voucher specimen for future reference.

Reply: Thank you for your comment. In the revised manuscript, we have included the collection sites of the plant in accordance with scientific standards, along with the GPS coordinates. Additionally, the collection habitat, coordinates, altitude, and collector number have all been clearly specified for future reference. We hope this meets your expectations.

3. Authors need to mention the proper rationale for the selection of extraction solvents and method of extraction (maceration). Authors also need to give the details of order of solvents used for the extraction.

Reply: Thank you for your comment. In the revised manuscript, we have provided a detailed rationale for the selection of extraction solvents and the method of extraction (maceration). The solvents were chosen based on their polarity to effectively extract a broad range of bioactive compounds. We started with the most polar solvent (aqueous) and proceeded to the least polar solvent (n-hexane) to ensure efficient separation of compounds according to their solubility. This sequential order maximizes the extraction of relevant bioactive compounds from the plant material.

4. Mention the reason for moistening of filter paper with sodium phosphate before filter paper.

Reply: Thank you for your comment. In the revised manuscript, we have clarified that before filtration, the filter paper was moistened with a solution of sodium phosphate and ethanol (95%) to prevent the likelihood of plant extract components, such as phenolics, flavonoids, or enzymes, from binding to the dry paper and being lost during filtration. This step ensures more efficient collection of the extract components.

5. Authors are suggested to check the chemical names/formulas and botanical names for typos.

Reply: Thank you for your suggestion. We have carefully reviewed the chemical names/formulas and botanical names in the manuscript and have corrected any typographical errors, grammar and text clarity changes have been highlighted with green color. We appreciate your attention to detail and have ensured accuracy in the revised version.

7. Table 2: Can glycosides and carbohydrates be detected in n-hexane extract?

Reply: Thank you for your comment. Upon reviewing the data again, we have confirmed that glycosides, carbohydrates, and leucoanthocyanins were not detected in the n-hexane extract. We have updated Table 2 to reflect this correction.

8. Justify the presence of carbohydrates and terpenoids in n-hexane and methanol extract but not in ethyl acetate extract during sequential extraction.

Reply: Thank you for your comment. The presence of carbohydrates and terpenoids in the n-hexane and methanol extracts, but not in the ethyl acetate extract, can be explained by the varying polarity of these compounds. Terpenoids, which are present in both n-hexane and methanol extracts, may include both highly polar and non-polar forms of terpenoids that are not easily detectable in the ethyl acetate extract. Ethyl acetate, being a moderately polar solvent, might not effectively extract certain types of terpenoids, particularly those with non-polar characteristics.

9. Table 3 and 4- Check the IC50 values of test extracts with reference to the percent inhibition. E.g. Table3- Methanolic extract gives 47.53 % inhibition at 500 microgrm/ml but IC50 value mentioned is 190.95. Justify.

Reply: Thank you for your careful review and for pointing out this inconsistency. We sincerely appreciate your efforts to highlight the mistake. Upon re-evaluation, we have corrected the IC50 values in the revised manuscript to accurately reflect the percent inhibition. We apologize for the oversight and thank you for bringing it to our attention.

10. Authors are recommended to discuss the in-vitro assays results with reference to the recent literature showing similar studies. Authors are suggested to read the following articles for better clarity;

Reply: Thank you for sharing these valuable references. We have reviewed the suggested literature and incorporated the relevant studies into both the Introduction and Discussion sections of the manuscript. This addition provides better context and comparison for our in-vitro assay results. We appreciate your helpful guidance in enhancing the clarity of our work.

https://doi.org/10.1016/j.jics.2022.100807

https://doi.org/10.1016/

---

## [Decision Letter · Decision Letter 1]

12 Apr 2025

PONE-D-24-51482R1Phytochemical Composition and Bioactivity of Debregeasia saenab  Leaves: Insights into Anti-Diabetic and Antioxidant PropertiesPLOS ONE

Dear Dr. Ahmad,

Thank you for submitting your manuscript to PLOS ONE. After careful consideration, we feel that it has merit but does not fully meet PLOS ONE’s publication criteria as it currently stands. Therefore, we invite you to submit a revised version of the manuscript that addresses the points raised during the review process.

We look forward to receiving your revised manuscript.

Kind regards,

Hamida Hamdi Mohammed Ismail, ph.D.

Academic Editor

PLOS ONE

Journal Requirements:

Additional Editor Comments :

-GC figure very bad, the authors should be improved.

Reviewers' comments:

Reviewer's Responses to Questions

**Comments to the Author**

1. If the authors have adequately addressed your comments raised in a previous round of review and you feel that this manuscript is now acceptable for publication, you may indicate that here to bypass the “Comments to the Author” section, enter your conflict of interest statement in the “Confidential to Editor” section, and submit your "Accept" recommendation.

Reviewer #2: All comments have been addressed

Reviewer #5: (No Response)

2. Is the manuscript technically sound, and do the data support the conclusions?

Reviewer #2: Yes

Reviewer #5: Yes

3. Has the statistical analysis been performed appropriately and rigorously? 

Reviewer #2: Yes

Reviewer #5: Yes

4. Have the authors made all data underlying the findings in their manuscript fully available?

Reviewer #2: Yes

Reviewer #5: Yes

5. Is the manuscript presented in an intelligible fashion and written in standard English?

Reviewer #2: Yes

Reviewer #5: Yes

6. Review Comments to the Author

Reviewer #2: The authors have revised the manuscript as per the suggestions and can be accepted in its present form.

Reviewer #5: Dear Editor; The attached article was checked. The manuscript contains interesting information about Phytochemical Composition and Bioactivity of Debregeasia saenab Leaves: Insights into Anti-Diabetic and Antioxidant Properties

I think that this article is well suited to your journal.

It is generally good work. The scientific and presentation level of the manuscript is high.

The title is understandable and in line with the text. The text is written in a descriptive and understandable language. The material and method are well described and adequately detailed Discussion and conclusion are interrelated.

In introduction and ms: Write the name of the authority to the end of the plant’s name.

- http://www.theplantlist.org/

Debregeasia saeneb (Forssk.) Hepper & J.R.I.Wood is an accepted name

Please, read the paper and correct them all.

What is the difference between the study and existing studies?

Some additions can be made to the key words. Phytochemical Composition; Bioactivity

References were cross-checked.

-The paper should be edited according to the writing rules of the journal

Original manuscript. There are, however, a few minor changes required.

7. PLOS authors have the option to publish the peer review history of their article (what does this mean? ). If published, this will include your full peer review and any attached files.

**Do you want your identity to be public for this peer review?** For information about this choice, including consent withdrawal, please see our Privacy Policy .

Reviewer #2: No

Reviewer #5: No

---

## [Author Response · Author response to Decision Letter 2]

2 Jun 2025

Response to Editor and Reviewers

Manuscript ID: PONE-D-24-51482R1

Title: Phytochemical Composition and Bioactivity of Debregeasia saeneb Leaves: Insights into Anti-Diabetic and Antioxidant Properties

Journal: PLOS ONE

Dear Editor Dr. Hamida Hamdi Mohammed Ismail

Thank you for your email and the opportunity to revise our manuscript. We sincerely appreciate the time and effort invested by the academic editor and reviewers in evaluating our work. We are grateful for their constructive feedback, which has helped us improve the quality and clarity of the manuscript.

We have carefully addressed all the comments and suggestions raised during the review process. Below, we provide a point-by-point response to each concern, detailing the revisions made in the manuscript. The changes have been incorporated into:

1. A marked-up copy with tracked changes (uploaded as Revised Manuscript with Track Changes).

2. A clean version of the revised manuscript (uploaded as Manuscript).

Editorial comments:

Reply: Thank you for your feedback. We have not changed financial disclosure. Additionally, we have make changes according to your instructions (While revising your submission, please upload your figure files to the Preflight Analysis and Conversion Engine (PACE) digital diagnostic tool, https://pacev2.apexcovantage.com/.) and uploaded the figures to the system

Journal Requirement

Comment:

Please review your reference list to ensure that it is complete and correct. If you have cited papers that have been retracted, please include the rationale for doing so in the manuscript text or remove these references and replace them with relevant current references. Any changes to the reference list should be mentioned in the rebuttal letter that accompanies your revised manuscript. If you need to cite a retracted article, indicate the article’s retracted status in the References list and also include a citation and full reference for the retraction notice.

Reply: Thank you for your comment regarding the reference list. We have carefully reviewed all the cited references and confirm that none of the papers included in our manuscript have been retracted. Therefore, no changes were necessary in the reference list. We appreciate your diligence in ensuring the integrity of the cited literature.

Additional Editor Comments:

-GC figure very bad, the authors should be improved.

Reply: Thank you for your feedback regarding the GC figure. We have improved the quality of the GC figure by enhancing its resolution and clarity to ensure better readability and presentation. We hope the revised figure now meets the journal’s standards.

Reviewers' comments:

Reviewer's Responses to Questions

Comments to the Author

1. If the authors have adequately addressed your comments raised in a previous round of review and you feel that this manuscript is now acceptable for publication, you may indicate that here to bypass the “Comments to the Author” section, enter your conflict-of-interest statement in the “Confidential to Editor” section, and submit your "Accept" recommendation.

Reviewer #2: All comments have been addressed

Reviewer #5: (No Response)

2. Is the manuscript technically sound, and does the data support the conclusions?

The manuscript must describe a technically sound piece of scientific research with data that supports the conclusions. Experiments must be conducted rigorously, with appropriate controls, replication, and sample sizes. The conclusions must be drawn appropriately based on the data presented.

Reviewer #2: Yes

Reviewer #5: Yes

3. Has the statistical analysis been performed appropriately and rigorously?

Reviewer #2: Yes

Reviewer #5: Yes

4. Have the authors made all data underlying the findings in their manuscript fully available?

The PLOS Data policy requires authors to make all data underlying the findings described in their manuscript fully available without restriction, with rare exception (please refer to the Data Availability Statement in the manuscript PDF file). The data should be provided as part of the manuscript or its supporting information or deposited to a public repository. For example, in addition to summary statistics, the data points behind means, medians and variance measures should be available. If there are restrictions on publicly sharing data—e.g. participant privacy or use of data from a third party—those must be specified.

Reviewer #2: Yes

Reviewer #5: Yes

5. Is the manuscript presented in an intelligible fashion and written in standard English?

Reviewer #2: Yes

Reviewer #5: Yes

6. Review Comments to the Author

Reviewer #2: The authors have revised the manuscript as per the suggestions and can be accepted in its present form.

Response: We are sincerely grateful to Reviewer #2 for their valuable time and insightful feedback on our manuscript. We are delighted that the reviewer found our revisions satisfactory and appreciate their positive assessment of our work. Their constructive suggestions during the initial review round significantly contributed to improving the quality of our manuscript.

Reviewer #5: Dear Editor; The attached article was checked. The manuscript contains interesting information about Phytochemical Composition and Bioactivity of Debregeasia saenab Leaves: Insights into Anti-Diabetic and Antioxidant Properties. I think that this article is well suited to your journal.

It is generally good work. The scientific and presentation level of the manuscript is high.

The title is understandable and in line with the text. The text is written in a descriptive and understandable language. The material and method are well described, and adequately detailed Discussion and conclusion are interrelated.

Comment: In introduction and ms: Write the name of the authority to the end of the plant’s name. - http://www.theplantlist.org/ Debregeasia saeneb (Forssk.) Hepper & J.R.I.Wood is an accepted name Please, read the paper and correct them all.

Reply: We are very thankful to the reviewer for their valuable comment. We have added authority to the end of the plant’s name in the revised manuscript

Comment: What is the difference between study and existing studies? Some additions can be made to the key words.

Reply: The study gap has been mentioned at the end of introduction (Line: 130-135). We have also added keywords.

Comment: References were cross-checked.

Reply: Thank you for confirming. We acknowledge that the references have been cross-checked.

Comment: -The paper should be edited according to the writing rules of the journal

Original manuscript. There are, however, a few minor changes required.

Reply: We have changed the whole manuscript according to the journal style.

7. PLOS authors have the option to publish the peer review history of their article (what does this mean?). If published, this will include your full peer review and any attached files.

If you choose “no”, your identity will remain anonymous, but your review may still be made public.

Do you want your identity to be public for this peer review? For information about this choice, including consent withdrawal, please see our Privacy Policy.

Reviewer #2: No

Reviewer #5: No

We believe these revisions have strengthened the manuscript and hope it now meets PLOS ONE’s publication criteria. Thank you again for your consideration. We look forward to your feedback.

---

## [Decision Letter · Decision Letter 2]

9 June 2025

Phytochemical Composition and Bioactivity of Debregeasia saenab  Leaves: Insights into Anti-Diabetic and Antioxidant Properties

PONE-D-24-51482R2

Dear Dr. Ahmad,

We’re pleased to inform you that your manuscript has been judged scientifically suitable for publication and will be formally accepted for publication once it meets all outstanding technical requirements.

Kind regards,

Hamida Hamdi Mohammed Ismail, ph.D.

Academic Editor

PLOS ONE

Additional Editor Comments (optional):

Reviewers' comments:

Reviewer's Responses to Questions

**Comments to the Author**

1. If the authors have adequately addressed your comments raised in a previous round of review and you feel that this manuscript is now acceptable for publication, you may indicate that here to bypass the “Comments to the Author” section, enter your conflict of interest statement in the “Confidential to Editor” section, and submit your "Accept" recommendation.

Reviewer #5: All comments have been addressed

2. Is the manuscript technically sound, and do the data support the conclusions?

Reviewer #5: Yes

3. Has the statistical analysis been performed appropriately and rigorously? 

Reviewer #5: (No Response)

4. Have the authors made all data underlying the findings in their manuscript fully available?

Reviewer #5: Yes

5. Is the manuscript presented in an intelligible fashion and written in standard English?

Reviewer #5: Yes

6. Review Comments to the Author

Reviewer #5: (No Response)

7. PLOS authors have the option to publish the peer review history of their article (what does this mean? ). If published, this will include your full peer review and any attached files.

**Do you want your identity to be public for this peer review?** For information about this choice, including consent withdrawal, please see our Privacy Policy .

Reviewer #5: No

---

## [Editor Report · Acceptance letter]

PONE-D-24-51482R2

PLOS ONE

Dear Dr. Ahmad,

I'm pleased to inform you that your manuscript has been deemed suitable for publication in PLOS ONE. Congratulations! Your manuscript is now being handed over to our production team.

Kind regards,

on behalf of

Professor Hamida Hamdi Mohammed Ismail

Academic Editor

PLOS ONE